# Brains and algorithms partially converge in natural language processing

Charlotte Caucheteux[1,2]✉ & Jean-Rémi King [1,3]✉

Deep learning algorithms trained to predict masked words from large amount of text have recently been shown to generate activations similar to those of the human brain. However, what drives this similarity remains currently unknown. Here, we systematically compare a variety of deep language models to identify the computational principles that lead them to generate brain-like representations of sentences. Specifically, we analyze the brain responses to 400 isolated sentences in a large cohort of 102 subjects, each recorded for two hours with functional magnetic resonance imaging (fMRI) and magnetoencephalography (MEG). We then test where and when each of these algorithms maps onto the brain responses. Finally, we estimate how the architecture, training, and performance of these models independently account for the generation of brain-like representations. Our analyses reveal two main findings. First, the similarity between the algorithms and the brain primarily depends on their ability to predict words from context. Second, this similarity reveals the rise and maintenance of perceptual, lexical, and compositional representations within each cortical region. Overall, this study shows that modern language algorithms partially converge towards brain-like solutions, and thus delineates a promising path to unravel the foundations of natural language processing.

---

[1] Facebook AI Research, Paris, France. [2] Université Paris-Saclay, Inria, CEA, Palaiseau, France. [3] École normale supérieure, PSL University, CNRS, Paris, France.
✉email: ccaucheteux@fb.com; jeanremi@fb.com

Deep learning algorithms have recently made considerable progress in developing abilities generally considered unique to the human species[1–3]. Language transformers, in particular, can complete, translate, and summarize texts with an unprecedented accuracy[4–7]. These advances raise a major question: do these algorithms process words and sentences like the human brain?

Recent neuroimaging studies suggest that they might—at least partially[8–12]. First, word embeddings—high dimensional dense vectors trained to predict lexical neighborhood[13–16]—have been shown to linearly map onto the brain responses elicited by words presented either in isolation[17–19] or within narratives[20–30]. Second, the contextualized activations of language transformers improve the precision of this mapping, especially in the pre-frontal, temporal and parietal cortices[31–33]. Third, specific computations of deep language models, such as the estimations of word surprisal (i.e., the probability of a word given its context) and the parsing of syntactic constituents have been shown to correlate with evoked related potentials[30,34–36] and functional magnetic resonance imaging (fMRI)[28,36]. However, the above studies remain fragmentary: first, most only analyze a small number of subjects (although see refs. [20,28,29]). Second, most studies only explore the spatial but not the temporal properties of the brain responses to language (although see refs. [30,33]).

More critically, the principles that lead a deep language models to generate brain-like representations remain largely unknown. Indeed, past studies only investigated a small set of pretrained language models that typically vary in dimensionality, architecture, training objective, and training corpus. The inherent correlations between these multiple factors thus prevent identifying those that lead algorithms to generate brain-like representations.

To address this issue, we systematically compare a wide variety of deep language models in light of human brain responses to sentences (Fig. 1). Specifically, we analyze the brain activity of 102 healthy adults, recorded with both fMRI and source-localized

magneto-encephalography (MEG). During these two 1 h-long sessions the subjects read isolated Dutch sentences composed of 9–15 words[37]. After quantifying the signal-to-noise ratio of the brain responses (Fig. 2), we train a variety of deep learning algorithms, extract their responses to the very same sentences and compare their ability to linearly map onto the fMRI and MEG brain recordings. Finally, we assess how the training, the architecture, and the word-prediction performance independently explains the brain-similarity of these algorithms and localize this convergence in both space and time.

We find that (1) a variety of deep learning algorithms linearly map onto the brain areas associated with reading (Fig. 3), (2) the best brain-mapping are obtained from the middle layers of deep language models and, critically, we show that (3) whether an algorithm maps onto the brain primarily depends on its ability to predict words context (Fig. 4).

## Results

**Shared brain responses to words and sentences across subjects.** Before comparing deep language models to brain activity, we first aim to identify the brain regions recruited during the reading of sentences. To this end, we (i) analyze the average fMRI and MEG responses to sentences across subjects and (ii) quantify the signal-to-noise ratio of these responses, at the single-trial single-voxel/sensor level.

As expected[38–41], the average fMRI and MEG responses to words reveals a hierarchy of neural responses originating in V1 around 100 ms and continuing within the left posterior fusiform gyrus around 200 ms, the superior and middle temporal gyri, as well as the pre-motor and infero-frontal cortices between 150 and 500 ms after word onset (Supplementary Movie 1 and Supplementary Note 1 and Fig. 2a).

To quantify the proportion of these brain responses that depend on the specific content of sentences, we fit, for each subject separately, a shared response model across subjects

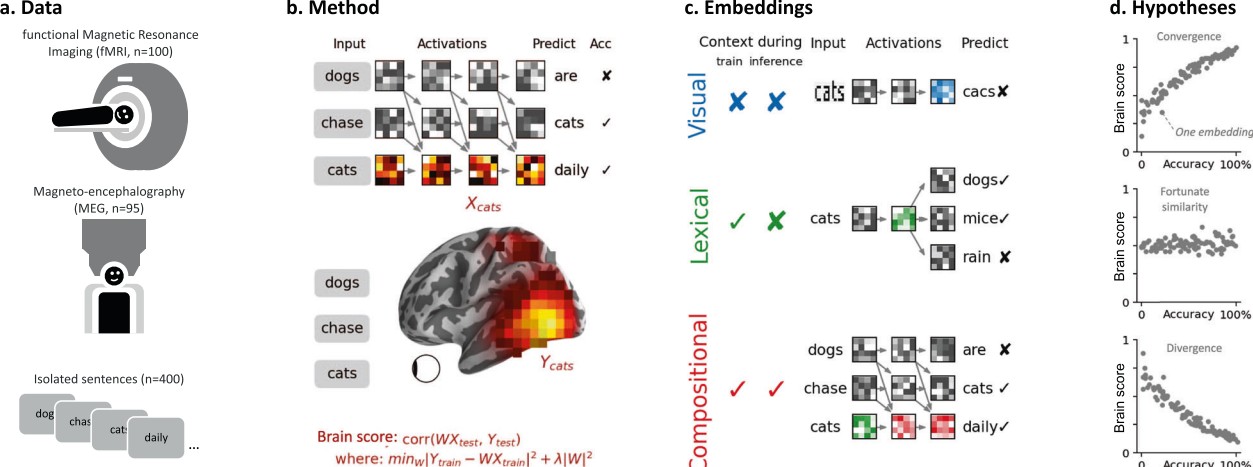

**Fig. 1 Approach. a** Subjects read isolated sentences while their brain activity was recorded with fMRI and MEG[37]. **b** To compute the similarity between a deep language model and the brain, we (1) fit a linear regression $W$ from the model's activations $X$ to predict brain responses $Y$ and (2) evaluate this mapping with a correlation between the predicted and true brain responses to held-out sentences $Y_{test}$. **c** We consider different types of embedding depending on whether they vary with neighboring words during training and/or during inference. Visual embeddings refer, here, to the activations of a deep convolutional neural network trained on character recognition. Lexical embeddings refer, here, to the non-contextualized activations associated with a word independently of its context. Here, we use the word-embedding layer of language transformers (bottom green), as opposed to algorithms like Word2Vec[93] (middle, green). Compositional embeddings refer, here, to the context-dependent activations of a deep language model (see SI.4 for a discussion of our terminology). **d** The three panels represent three hypotheses on the link between deep language models and the brain. Each dot represents one embedding. Algorithm are said to *converge* to brain-like computations if their performance (*x*-axis: i.e., accuracy at predicting a word from its previous context) indexes their ability to map onto brain responses to the same stimuli (i.e., *y*-axis: brain score). High-dimensional neural networks can, in principle, capture relevant information[94,95] and thus lead to a fortunate similarity with brain responses, and event a systematic divergence.

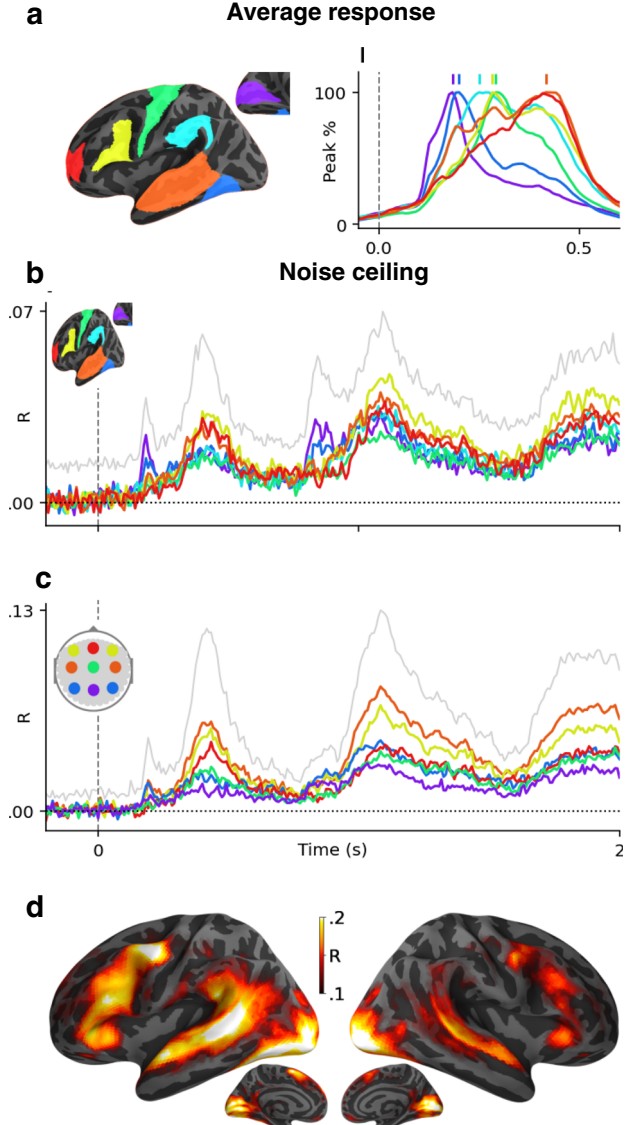

**Fig. 2 Average and shared response modeling (or noise ceiling). a** Grand average MEG source estimates to word onset ($t = 0$ ms) for seven regions typically associated with reading (V1: purple, M1: green, fusiform gyrus: dark blue, supramarginal gyrus: light blue, superior temporal gyrus: orange, infero-frontal gyrus: yellow and fronto-polar gyrus: red), normalized to their peak response. Vertical bars indicate the peak time of each region. The full (not normalized) spatio-temporal time course of the whole-brain activity is displayed in Supplementary Movie 1. **b** MEG shared response model (or noise ceilings), approximated by predicting brain responses of a given subject from those of all other subjects. Colored lines depict the mean noise ceiling in each region of interest. The gray line depicts the best noise ceiling across sources. **c** Same as **b** in sensor space. **d** Shared response model of fMRI recordings.

(or noise-ceiling, see "Methods" section, Supplementary Note 2 and Supplementary Table 1 and Fig. 2b–d). We then assess the accuracy of this model with a Pearson $R$ correlation (hereafter referred to as "brain score" following[42]) between the true and the predicted brain responses to held-out sentences, using a five-fold cross-validation. Finally, we assess the statistical significance of these brain scores with a two-sided Wilcoxon test across subjects, after testing for multiple comparison using false discovery rate (FDR) across voxels (see "Methods" section). Our shared response model confirms that the brain network classically

associated with language processing elicits representations specific to words and sentences[17,43,44].

**Deep language models reveal the hierarchical generation of language representations in the brain.** Where and when are the language representations of the brain similar to those of deep language models? To address this issue, we extract the activations ($X$) of a visual, a word and a compositional embedding (Fig. 1d) and evaluate the extent to which each of them maps onto the brain responses ($Y$) to the same stimuli. To this end, we fit, for each subject independently, an $\ell_2$-penalized regression ($W$) to predict single-sample fMRI and MEG responses for each voxel/sensor independently. We then assess the accuracy of this mapping with a brain-score similar to the one used to evaluate the shared response model.

Overall, the brain scores of these trained models are largely above chance (all $p < 10^{-9}$, Fig. 4a, e). The modest correlation values are consistent with the high level of noise in single-sample single-voxel/channel neuroimaging data (Fig. 2b–d). For example, fMRI and MEG scores reach $R = 0.048$ and $R = 0.041$, respectively, for the compositional embedding, which is close to and even exceeds our shared response model (fMRI: $R = 0.060$, MEG: $R = 0.020$, Fig. 2).

In fMRI, the brain scores of the visual embedding peak in the early visual cortex (V1) (mean brain scores across voxels: $R = 0.022 \pm 0.003$, $p < 10^{-11}$). By contrast, the brain scores of lexical embedding peak in the left superior temporal gyrus ($R = 0.052 \pm 0.004$, $p < 10^{-13}$) as well as in the inferior temporal cortex and middle frontal gyrus ($R = 0.053 \pm 0.003$, $p < 10^{-15}$) and are significant across the entire language and reading network (Fig. 3b). Finally, the brain scores of the compositional embedding are significantly higher than those of lexical of embeddings in the superior temporal gyrus ($\Delta R = 0.012 \pm 0.001$, $p < 10^{-16}$), the angular gyrus ($\Delta R = 0.010 \pm 0.001$, $p < 10^{-16}$), the infero-frontal cortex ($\Delta R = 0.016 \pm 0.001$, $p < 10^{-16}$) and the dorsolateral prefrontal cortex ($\Delta R = 0.012 \pm 0.001$, $p < 10^{-13}$). While these effects are lateralized (left hemisphere versus right hemisphere: $\Delta R = 0.010 \pm 0.001$, $p < 10^{-14}$), they are significant across a remarkably large number of bilateral areas (Fig. 3b). Lexical and compositional embeddings accurately predict brain responses in the early visual cortex. This result is not necessarily surprising: language embeddings encode features (e.g., position of words in the sentence, beginning/end of the sentence) that correlate with visual information (words are flashed at a screen, and the sentences are separated by pauses). Critically, the gain ($\Delta R$) of these embeddings remain very small, suggesting that this effect is mainly driven by the covariance between low-level and high-level representations of words.

**Tracking the sequential generation of language representations over time and space.** To characterize the dynamics of these brain representations, we perform the same analysis using source-localized MEG recordings. The resulting brain scores are consistent with—although less spatially precise than—the above fMRI results (Fig. 3c, average brain score between 0 and 2 s). For clarity, Fig. 3d and Supplementary Movie 2 plot the gain in MEG scores: i.e., the difference of prediction performance between i) word and visual embeddings (green) and ii) the difference between compositional and word embedding (red). The brain scores of the visual embedding peak around 100 ms in V1 ($R = 0.008 \pm 0.002$, $p < 10^{-3}$), and rapidly propagate to higher-level areas (Fig. 3d and Supplementary Movie 2). The gain achieved by the word embedding can be observed in the left posterior fusiform gyrus around 200 ms and peaks around 400 ms and in the left temporal and frontal cortices. Finally, the gain

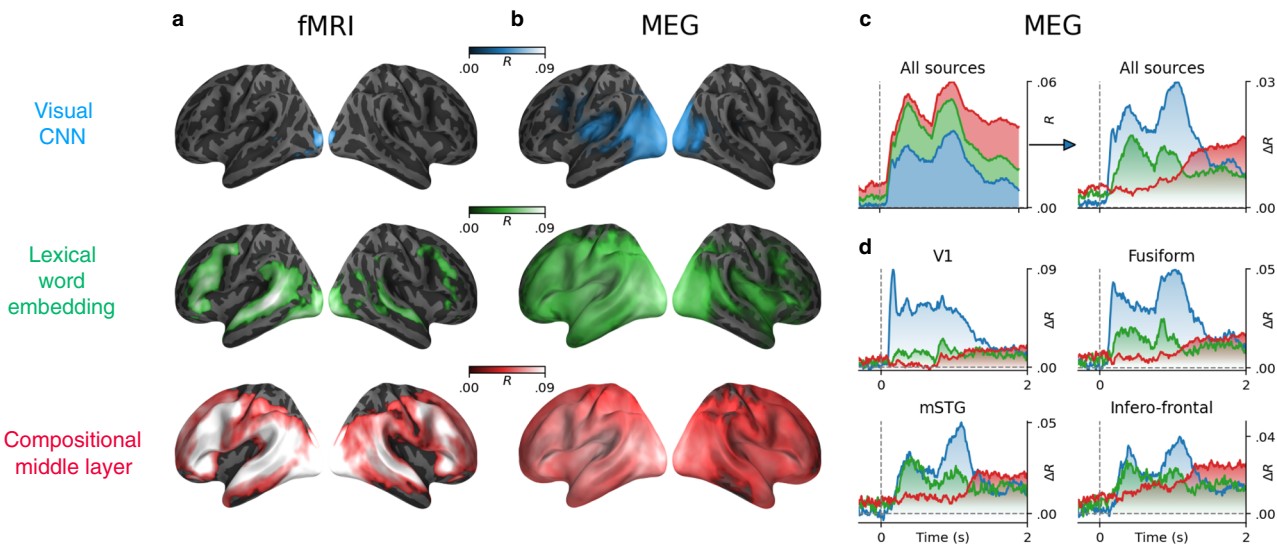

**Fig. 3 Brain-score comparison across embeddings.** Lexical and compositional representations (see Supplementary Note 4 for the definition of compositionality) can be isolated from (i) the word embedding layer (green) and (ii) one middle layer (red) of a typical language transformer (here, the ninth layer of a 12-layer causal transformer), respectively. We also report the brain scores of a convolutional neural network trained on visual character recognition (blue) to account for low-level visual representations. **a** Mean (across subjects) fMRI scores obtained with the visual, word, and compositional embeddings. All colored regions display significant fMRI scores across subjects ($n = 100$) after false discovery rate (FDR) correction. **b** Mean MEG scores averaged across all time samples and subjects ($n = 95$ subjects). **c** Left: mean MEG scores averaged across all sensors. Right: mean MEG gains averaged across all sensors: i.e., the gain in MEG score of one level relative to the level below (blue: $R$[visual]; green: $R$[word] $-$ $R$[visual]; red: $R$[compositional] $-$ $R$[word]). **d** Mean MEG gains in four regions of interest. For a whole-brain depiction of the MEG gains, see Supplementary Movie 2. For the raw scores (without subtraction), see Supplementary Fig. 6. For the distribution of scores across channels and voxels, see Supplementary Fig. 4.

achieved by the compositional embedding is observed in a large number of bilateral brain regions, and peaks around 1 s after word onset (Fig. 3c, d).

After that period, brain areas outside the language network, such as area V1, appear to be better predicted by word and compositional embeddings than by visual ones (e.g., between visual and word in V1: $\Delta R = 0.016 \pm 0.002$, $p < 10^{-10}$). These effects could thus reflect feedback activity[45] and explain why the corresponding fMRI responses are better accounted for by word and compositional embeddings than by visual ones.

Together with Supplementary Fig. 1, these results show with unprecedented spatio-temporal precision, that the brain-mapping of our three representative embeddings automatically recovers the hierarchy of visual, lexical, and compositional representations of language in each cortical region.

**Compositional embeddings best predict brain responses.** What computational principle leads these deep language models to generate brain-like activations? To address this issue, we generalize the above analyses and evaluate the brain scores of 36 transformer architectures (varying from 4 to 12 layers, each ranging from 128 to 512 dimensions, and each benefiting from 4 to 8 attention heads), trained on the same Wikipedia dataset either with a causal language modeling (CLM) or a masked language modeling task (MLM). While causal language models are trained to predict a word from its previous context, masked language models are trained to predict a randomly masked word from its both left and right context.

Overall, we observe that the corresponding brain scores largely vary as a function of the relative depth of the embedding within the language transformer. Specifically, both MEG and fMRI scores follow an inverted U-shaped pattern across layers for all architectures (Fig. 4a, e): the middle layers systematically outperform the output (fMRI: $\Delta R = 0.011 \pm 0.001$, $p < 10^{-18}$, MEG: $\Delta R = 0.003 \pm 0.0005$, $p < 10^{-13}$) and the input layers

(fMRI: $\Delta R = .031 \pm .001$, $p < 10^{-18}$, MEG: $\Delta R = .009 \pm .001$, $p < 10^{-17}$). For simplicity, we refer to "middle layers" as the layers $l \in [n_{layers}/2, 3n_{layers}/4]$ in Fig. 4a, e. This result confirms that the intermediary representations of deep language transformers are more brain-like than those of the input and output layers[33].

**The emergence of brain-like representations predominantly depends on the algorithm's ability to predict missing words.** The above findings result from trained neural networks. However, recent studies suggest that random (i.e., untrained) networks can significantly map onto brain responses[27,46,47]. To test whether brain mapping specifically and systematically depends on the language proficiency of the model, we assess the brain scores of each of the 32 architectures trained with 100 distinct amounts of data. For each of these training steps, we compute the top-1 accuracy of the model at predicting masked or incoming words from their contexts. This analysis results in 32,400 embeddings, whose brain scores can be evaluated as a function of language performance, i.e., the ability to predict words from context (Fig. 4b, f).

We observe three main findings. First, random embeddings systematically lead to significant brain scores across subjects and architectures. The mean fMRI score across voxels is $R = 0.019 \pm 0.001$, $p < 10^{-16}$. The mean MEG score across channels and time sample is $R = 0.018 \pm 0.0008$, $p < 10^{-16}$. This result suggests that language transformers partially map onto brain responses independently of their language abilities.

Second, brain scores strongly correlate with language accuracy in both MEG ($R = 0.77$ Pearson's correlation on average $\pm 0.01$ across subjects) and fMRI ($R = 0.57 \pm 0.02$, Fig. 4b, c). The correlation is highest for middle (fMRI: $R = 0.81 \pm 0.02$; MEG: $R = 0.86 \pm 0.01$) than input (fMRI: $R = 0.39 \pm 0.03$; MEG: $R = 0.73 \pm 0.02$) and output layers (fMRI: $R = 0.63 \pm 0.03$; MEG: $R = 0.78 \pm 0.02$). Beta coefficients for each particular layer

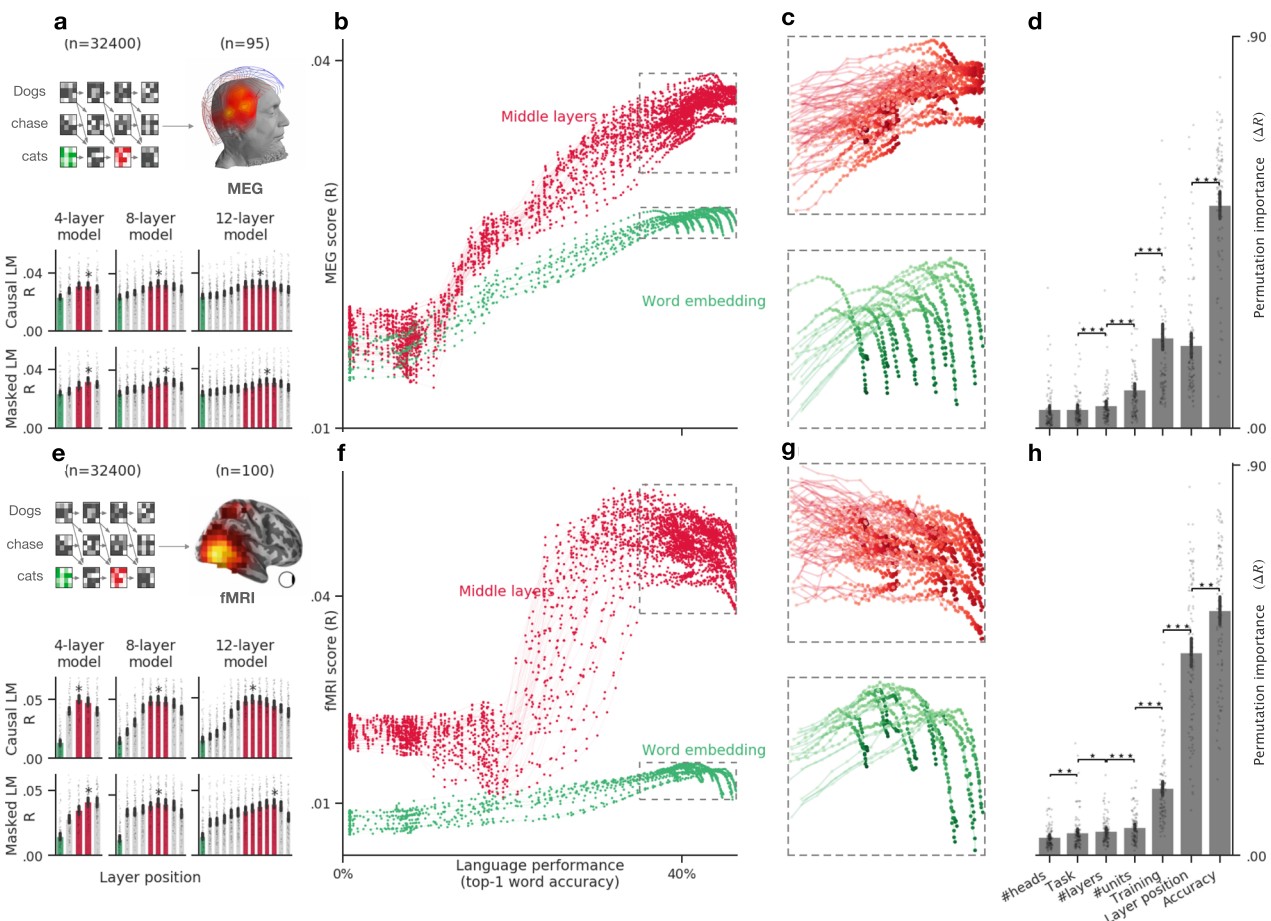

**Fig. 4 Language transformers tend to converge towards brain-like representations. a** Bar plots display the average MEG score (across time and channels) of six representative transformers varying in tasks (causal vs. masked language modeling) and depth (4–12 layers). The green and red bars correspond to the word-embedding and middle layers, respectively. The star indicates the layer with the highest MEG score. **b** Average MEG scores (across subjects, time, and channels) of each of the embeddings (dots) extracted from 18 causal architectures, separately for the input layer (word embedding, green) and the middle layers (red). **c** Zoom of **b**, focusing on the best neural networks (i.e., word-prediction accuracy >35%). The results reveal a plateau and/or a divergence of the middle and input layers. **d** Permutation importance quantifies the extent to which each property of the language transformers specifically contribute to making its embeddings more-or-less similar to brain activity ($\Delta R$). All properties (training task. dimensionality *etc.*) significantly contribute to the brain scores ($\Delta R > 0$, all $p < 0.0001$ across subjects). Ordered pairwise comparisons of the permutation scores are marked with a star ($*p < 0.05$, $**p < 0.01$, $***p < 0.001$). **e**–**h** Same as **a**–**d**, but evaluated on fMRI recordings. All error bars are the 95% confidence intervals across subjects ($n = 95$ for MEG, $n = 100$ for fMRI).

and architecture are displayed in Supplementary Fig. 1a, b. Furthermore, single-voxel analyses show that this correlation between brain score and language performance is driven mainly by the superior temporal sulcus and gyrus for the embedding layer (mean $R = 0.52 \pm 0.06$) and is widespread for the middle layers, exceeding a correlation of $R = 0.85$ in the superior temporal sulcus, infero-frontal, fusiform and angular gyri (Supplementary Fig. 1c). Overall, this result suggests that the better language models are at predicting words from context, the more their activations linearly map onto those of the brain.

Third, the highest brain scores are not achieved by the very best language transformers (Fig. 4c, g). For instance, CLM transformers best map onto MEG ($R = 0.039$) and fMRI ($R = 0.056$) when they reach a language performance of 43% and 32%, respectively. By contrast, the very best transformers reach a language accuracy of 46%, but have significantly smaller brain scores (Fig. 4c, g).

**Architectural and training factors impact brain scores too.** Language performance co-varies with the amount of training as

well as with several architectural variables. To disentangle the contribution of each of these variables to the brain scores, we perform a permutation feature importance analysis. Specifically, we train a Random Forest estimator[48] to predict the average brain scores (across voxels or MEG sensors) of each subject independently, given the layer of the representation, the architectural properties (number of layers, dimensionality, and attention head), task (CLM and MLM), amount of training (number of steps) and language performance (top-1 accuracy) of the transformer. Permutation feature importance then estimates the unique contribution of each feature in explaining the variability of brain scores across models[48,49]. The results confirm that language performance is the most important factor that drives brain scores (Fig. 4d–h). This factor supersedes other covarying factors such as the amount of training, and the relative position of the embedding with regard to the architecture ("layer position"): $\Delta R = 0.56 \pm 0.01$ for fMRI, $\Delta R = 0.51 \pm 0.02$ for MEG. Nevertheless, these other factors contribute significantly to the prediction of brain scores ($p < 10^{-16}$ across subjects for all variables).

Overall, these results show that the ability of deep language models to map onto the brain primarily depends on their ability to predict words from the context, and is best supported by the representations of their middle layers.

## Discussion

Do deep language models and the human brain process sentences in the same way? Following a recent methodology[33,42,44,46,46,50–56], we address this issue by evaluating whether the activations of a large variety of deep language models linearly map onto those of 102 human brains. Our study provides two main contributions.

First, our work complements previous studies[26,27,30–34] and confirms that the activations of deep language models significantly map onto the brain responses to written sentences (Fig. 3). This mapping peaks in a distributed and bilateral brain network (Fig. 3a, b) and is best estimated by the middle layers of language transformers (Fig. 4a, e). The notion of representation underlying this mapping is formally defined as linearly-readable information. This operational definition helps identify brain responses that any neuron can differentiate—as opposed to entangled information, which would necessitate several layers before being usable[57–61].

Furthermore, the comparison between visual, lexical, and compositional embeddings precise the nature and dynamics of these cortical representations. In particular, our results shows with unprecedented spatio-temporal precision that early visual responses (<150 ms) are quasi-entirely accounted for by visual embeddings, and then transmitted to the posterior fusiform gyrus, which switches from visual to lexical representations around 200 ms (Movie 2). This finding strengthens the claim that this area is responsible for orthographic and morphemic computations[39,62,63]. Then, around 400 ms, word embeddings predict a large fronto-temporo-parietal network which peaks in the left temporal gyrus; these word representations are then maintained for several seconds[17,19,31,33]. This result not only confirms the wide spread distribution of meaning in the brain[44,64], but also reveals its remarkably long-lasting nature.

Finally, compositional embeddings peak in the brain regions associated with high-level language processing such as the inferofrontal and the anterior temporal cortices as well as the superior temporal cortex and the temporal-parietal junction[35,41,65]. We confirm that these left-lateralized representations are significant in both hemispheres[66,67]. Critically, MEG suggests that these compositional effects become dominant and clearly bilateral long after word onset (>800 ms). We speculate that this surprisingly late responses may be due to the complexity of the sentences used in the present study, which may slow down compositional computations.

At this stage, however, these three levels representations remain coarsely defined. Further inspection of artificial[8,68] and biological networks[10,28,69] remains necessary to further decompose them into interpretable features. In particular, it will be important to test whether the converging representations presently identified solely correspond to well-known linguistics phenomena as our supplementary analyses suggest (Supplementary Fig. 2 and Supplementary Note 3), or, on the contrary, whether they correspond to unknown language structures.

Second, our study shows that the similarity between deep language models and the brain primarily depends on their ability to predict words from their context. Specifically, we show that language performance is the most contributing factor explaining the variability of brain scores across embeddings (Fig. 4d, h). Analogous results have been reported in both vision and audition research, where best deep learning models tend to best map onto brain responses[27,42,55,70,71]. In addition, our results are consistent

with the findings of Schrimpf et al.[27] reported simultaneously to ours. Together, these results suggest that deep learning algorithms converge—at least partially—to brain-like representations during their training. This result is not trivial: the representations that are optimal to predict masked or future words from large amounts of text could have been very distinct from those the brain learns to generate.

The mapping between deep language models and brain recordings reaches very low correlation values. This phenomenon is expected: i) neuroimaging is notoriously noisy and ii) we analyze and model here single-sample responses of single-voxel/sensor. However, the resulting brain scores are i) highly significant (all $p < 10^{-9}$ on average across both all fMRI voxels and MEG sensors), including when compared to a permutation baseline (Supplementary Fig. 3), and ii) in the same order of magnitude than a baseline shared-response model (or noise ceiling, Fig. 2) as well as previous reports (see e.g., [44] before correcting for the noise ceiling). Besides, we generally report brain scores averaged across all voxels or MEG channels, even though many brain areas do not strongly respond to language (Fig. 2). Critically, the link between brain scores and language performance is strong: the correlation between the language performance and brain scores is above $R = 0.90$ for MEG and $R = 0.80$ for fMRI (Supplementary Fig. 1). Nevertheless, it is clear that improving the the signal-to-noise ratio, for instance by using increasingly large datasets[20,29,47,72] will be critical to precisely characterize the nature of brain representations.

Permutation feature importance shows that several factors such as the amount of training and the architecture significantly impact brain scores. This finding contributes to a growing list of variables that lead deep language models to behave more-or-less similarly to the brain. For example, Hale et al.[36] showed that the amount and the type of corpus impact the ability of deep language parsers to linearly correlate with EEG responses. The present work complements this finding by evaluating the full set of activations of deep language models. It further demonstrates that the key ingredient to make a model more brain-like is, for now, to improve its language performance.

The conclusion that deep networks converge towards brain-like representations should be qualified: we show that the brain scores of the very best models tend to ultimately decrease with language performance, especially in fMRI (Fig. 4g). We speculate that this phenomenon (also observed in vision[70]) may rise because transformers overfit an inappropriate objective. Specifically, while there is growing evidence that the human brain does predict words from context[30,73,74], this learning rule may not fully account for the complex (and potentially various) tasks performed by the brain (e.g., long-range[75,76] and hierarchical predictions[77]).

This discrepancy adds to the long-list of differences between deep language models and the brain: whereas the brain is trained (i) with a recurrent architecture and (ii) on a relatively small amount of grounded sentences, transformers are trained (i) with a massively feedforward architecture and (ii) on huge text databases[7] (note that, given large-enough spaces, feedforward transformers may actually implement computations similar to recurrent networks[78]). Consequently, while the similarity between deep networks and the brain provide a stepping stone to unravel the foundation of natural language processing, identifying the remaining differences between these two systems remains, by far, the major challenge to build algorithms that learn and think like humans[7,9,79,80].

## Methods

**Deep language transformers**. To model word and sentence representations, we trained a variety of transformers[4], and input them with the same sentences that the

subject read. Transformers consist of multiple contextual transformer layers stacked onto one non-contextualized word embedding layer (a look-up table). Following the standard implementation[4,81,82], the word embedding layer is trained simultaneously with the contextual layers: the weights of the word embedding vary with the training, and so do their activations in response to fixed inputs. Thus, one representation can be extracted from each (contextual or non-contextual) layer. We always extract the activations in a causal way: for example, given the sentence "THE CAT IS ON THE MAT", the brain response to "ON" would be solely compared to the activations of the transformer input with "THE CAT IS ON", and extracted from the "ON" contextualized embeddings. Word embeddings and contextualized embeddings were generated for every word, by generating word sequences from the three previous sentences. We did not observe qualitatively different results when using shorter or longer contexts. It is to be noted that the sentences were isolated, and were not part of a narrative.

In total, we investigated 32 distinct architectures varying in their dimensionality ($\in [128, 256, 512]$), number of layers ($\in [4, 8, 12]$), attention heads ($\in [4, 8]$), and training task (causal language modeling and masked language modeling). While causal language transformers are trained to predict a word from its previous context, masked language transformers predict randomly masked words from a surrounding context. We froze the networks at ≈100 training stages (log distributed between 0 and 4, 5 M gradient updates, which corresponds to ≈35 passes over the full corpus), resulting in 3600 networks in total, and 32,400 word representations (one per layer). The training was early-stopped when the networks' performance did not improve after five epochs on a validation set. Therefore, the number of frozen steps varied between 96 and 103 depending on the training length.

The algorithms were trained using XLM implementation[6]. No hyper-parameter tuning was performed. Following [6], each algorithm was trained on eight GPUs using early stopping with training perplexity criteria, 16 streams per batch, 128 words per stream, epoch size of 200,000 streams, 0.1 dropout, 0.1 attention dropout, gelu activation, inverse (sqrt) adam optimizer with learning rate 0.0001, 0.01 weight decay, on the same Wikipedia corpus of 278,386,651 words (in Dutch) extracted using WikiExtractor[83] and pre-processed using Moses tokenizer[84], with punctuation. We restricted the vocabulary to the 50,000 most frequent words, concatenated with all words used in the study (50,341 vocabulary words in total). These design choices enforce that the difference in brain scores observed across models cannot be explained by differences in corpora and text preprocessing.

To evaluate the language processing performance of the networks, we computed their performance (top-1 accuracy on word prediction given the context) using a test dataset of 180,883 words from Dutch Wikipedia. The list of architectures and their final performance at next-word prerdiction is provided in Supplementary Table 2.

For clarity, we dissociate:

- The architectures (e.g., one transformer with 12 layers): there are 36 transformer architectures here (18 CLM and 18 MLM).
- The models: one architecture, frozen at one particular learning step. Since we use 100 learning steps, there are $36 \times 100 = 3600$ networks here.
- The embeddings: one word representation extracted from a network, at one particular layer. Since the number of layers varies with the architecture (twelve networks with 5, twelve networks with 9 and twelve networks with 13 twelve layers, including the non contextualized word embedding), there are $12 \times (5 + 9 + 13) = 324$ representations per step, so $324 \times 100 = 3400$ word embeddings in total.

**Visual convolutional neural network**. To model visual representations, every word presented to the subjects was rendered on a gray $100 \times 32$ pixel background with a centered black Arial font, and input to a VGG network pretrained to recognize words from images[85], resulting in an 888-dimensional embedding. Specifically, this model was trained on real pictures of single words taken in naturalistic settings (e.g., ad, banner).

This embedding was used to replicate and extend previous work on the similarity between visual neural network activations and brain responses to the same images (e.g., [42,52,53]).

**Neuroimaging protocol**. For all the analyses, we used the open-source dataset released by Schoffelen and colleagues[37], gathering the functional magnetic resonance imaging (fMRI) and magneto-encephalography (MEG) recordings of 204 native Dutch speakers (100 males), aged from 18 to 33 years. Here, we focused on the 102 right-handed speakers who performed a reading task while being recorded by a CTF magneto-encephalography (MEG) and, in a separate session, with a SIEMENS Trio 3T Magnetic Resonance scanner[37].

Words (in Dutch) were flashed one at a time with a mean duration of 351 ms (ranging from 300 to 1400 ms), separated with a 300 ms blank screen, and grouped into sequences of 9–15 words, for a total of approximately 2700 words per subject. Sequences were separated by a 5 s-long blank screen. We restricted our study to meaningful sentences (400 distinct sentences in total, 120 per subject). The exact syntactic structures of sentences varied across all sentences. Roughly, sentences were either composed of a main clause and a simple subordinate clause, or

contained a relative clause. Twenty percent of the sentences were followed by a yes/no question (e.g., "Did grandma give a cookie to the girl?") to ensure that subjects were paying attention. Questions were not included in the dataset, and thus excluded from our analyses. Sentences were grouped into blocks of five sequences. This grouping was used for cross-validation to avoid information leakage between the train and test sets.

**Magnetic resonance imaging (MRI)**. Structural images were acquired with a T1-weighted magnetization-prepared rapid gradient-echo (MP-RAGE) pulse sequence. The full acquisition details, available in ref. [37], are summarized here simplicity: TR = 2300 ms, TE = 3.03 ms, 8 degree flip-angle, 1 slab, slice-matrix size = $256 \times 256$, slice thickness = 1 mm, field of view = 256 mm, isotropic voxel-size = $1.0 \times 1.0 \times 1.0$ mm. Structural images were defaced by Schoffelen and colleagues. Preprocessing of the structural MRI was performed with Freesurfer[86], using the recon-all pipeline and a manual inspection of the cortical segmentations, realigned to "fsaverage". Region-of-interest analyses were selected from the PALS Brodmann's Area atlas[87] and the Destrieux atlas[88].

Functional images were acquired with a T2\*-weighted functional echo-planar blood oxygenation level-dependent (EPI-BOLD) sequence. The full acquisition details, available in ref. [37], are summarized here for simplicity: TR = 2.0 s, TE = 35 ms, flip angle = 90 degrees, anisotropic voxel size = $3.5 \times 3.5 \times 3.0$ mm extracted from 29 oblique slices. fMRI was preprocessed with fMRIPrep with default parameters[89]. The resulting BOLD times series were detrended and de-confounded from 18 variables (the six estimated head-motion parameters ($trans_{x,y,z}$ and $rot_{x,y,z}$) and the first six noise components calculated using anatomical CompCorr[90] and six DCT-basis regressors using nilearn's clean_img pipeline and otherwise default parameters[91]. The resulting volumetric data lying along a 3 mm line orthogonal to the mid-thickness surface were linearly projected to the corresponding vertices. The resulting surface projections were spatially decimated by 10, and are hereafter referred to as voxels, for simplicity. Finally, each group of five sentences was separately and linearly detrended. It is noteworthy that our cross-validation never splits such groups of five consecutive sentences between the train and test sets. Two subjects were excluded from the fMRI analyses because of difficulties in processing the metadata, resulting in 100 fMRI subjects.

**Magneto-encephalography (MEG)**. The MEG time series were preprocessed using MNE-Python and its default parameters except when specified[92]. Signals were band-passed filtered between 0.1 and 40 Hz filtered, spatially corrected with a Maxwell Filter, clipped between the 0.01st and 99.99th percentiles, segmented between $-500$ ms to $+2000$ ms relative to word onset and baseline-corrected before $t = 0$. Reference channels and non-MEG channels were excluded from subsequent analyses, leading to 273 MEG channels per subject. We manually co-referenced (i) the skull segmentation of subjects' anatomical MRI with (ii) the head markers digitized before MEG acquisition. A single-layer forward model was generated with the Freesurfer-wrapper implemented in MNE-Python[92]. Due to the lack of empty-room recordings, the noise covariance matrix used for the inverse operator was estimated from the zero-centered 200 ms of baseline MEG activity preceding word onset. Subjects' source space inverse operators were computed using a dSPRM. The average brain responses displayed in Fig. 1d were computed as the square of the average evoked related field across all words for each subject separately, averaged across subjects, and finally divided by their respective maxima, to highlight temporal differences. Supplementary Movie 1 displays the average sources without normalization. Seven subjects were excluded from the MEG analyses because of difficulties in processing the metadata, resulting in 92 usable MEG recordings.

**Shared response model: Brain → Brain mapping**. To estimate the amount of explainable signal in each MEG and fMRI recording, we trained and evaluated, through cross-validation, a linear mapping model $W$ to predict the brain responses of a given subject to each sentence $Y$ from the aggregated brain responses of all other subjects who read the same sentence $X$. Specifically, five cross-validation splits were implemented across 5-sentence blocks with scikit-learn GroupKFold[49]. For each word of each sentence $i$, all but one subject who read the corresponding sentence were averaged with one another to form a template brain response: $x_i \in \mathbb{R}^n$ with $n$ the number of MEG channels or fMRI voxels, as well as a target brain response $y_i \in \mathbb{R}^n$ corresponding to the remaining subject. $X$ and $Y$ were normalized (mean = 0, std = 1) across sentences for each spatio-temporal dimension, using a robust scaler clipping below and above the 0.01st and 99.99th percentiles, respectively. A linear mapping $W \in \mathbb{R}^{n \times n}$ was then fit with a ridge regression to best predict $Y$ from $X$ on the train set:

$$W = (X_{train}^T X_{train} + \lambda I)^{-1} X_{train}^T Y_{train}, \tag{1}$$

with $\lambda$ the $l2$ regularization parameter, chosen amongst 20 values log-spaced between $10^{-3}$ and $10^8$ with nested leave-one-out cross-validation for each dimension separately (as implemented in ref. [49]). Brain predictions $\hat{Y} = WX$ were evaluated with a Pearson correlation on the test set:

$$R = \mathrm{Corr}(Y_{test}, \hat{Y}_{test}). \tag{2}$$

For the MEG source noise estimate, the correlation was also performed after source projection:

$$R = \text{Corr}(KY_{\text{test}}, K\hat{Y}_{\text{test}}) \qquad (3)$$

with $K \in \mathbb{R}^{n \times m}$ the inverse operator projecting the $n$ MEG sensors onto $m$ sources. Correlation scores were finally averaged across cross-validation splits for each subject, resulting in one correlation score ("brain score") per voxel (or per MEG sensor/time sample) per subject.

**Brain score and similarity: Network → Brain mapping**. To estimate the functional similarity between each artificial neural network and each brain, we followed the same analytical pipeline used for noise ceiling, but replaced $X$ with the activations of the deep learning models. Specifically, using the same cross-validation, and for each subject separately, we trained a linear mapping $W \in \mathbb{R}^{o,n}$ with $o$ the number of activations, to predict brain responses $Y$ from the network activations $X$. $X$ was normalized across words (mean = 0, std = 1).

To account for the hemodynamic delay between word onset and the BOLD response recorded in fMRI, we used a finite impulse response (FIR) model with five delays (from 2 to 10 s) to build $X^*$ from $X$. $W$ was found using the same ridge regression described above, and evaluated with the same correlation scoring procedure. The resulting brain correlation scores measure the linear relationship between the brain signals of one subject (measured either by MEG or fMRI) and the activations of one artificial neural network (e.g., a word embedding). For MEG, we simply fit and evaluated the model activations $X$ at each time sample independently.

In principle, one may orthogonalize low-level representations (e.g., visual features) from high-level network models (e.g., language model), to separate the specific contribution of each type of model. This is because middle layers have access to the word-embedding layer, and can, in principle, simply copy some of its activations. Similarly, word embedding can implicitly contain visual information: e.g., frequent words tend to be visually smaller than rare ones. In our case, however, the middle layers of transformers were much better than word embeddings, which were much better than visual embeddings. To quantify the gain $\Delta R$ achieved by a higher-level model $M_1$ (e.g., the middle layers of a transformer) and a lower level model $M_2$ (e.g., a word embedding) we thus simply compared the difference of their encoding scores:

$$\Delta R_{M_1} = R_{M_1} - R_{M_2} \qquad (4)$$

Results are consistent when using different orthogonalization methods (Supplementary Fig. 5).

**Convergence analysis**. All neural networks but the visual CNN were trained from scratch on the same corpus (as detailed in the first "Methods" section). We systematically computed the brain scores of their activations on each subject, sensor (and time sample in the case of MEG) independently. For computational reasons, we restricted model comparison on MEG encoding scores to ten time samples regularly distributed between [0, 2]s. Brain scores were then averaged across spatial dimensions (i.e., MEG channels or fMRI surface voxels), time samples, and subjects to obtain the results in Fig. 4. To evaluate the convergence of a model, we computed, for each subject separately, the correlation between (1) the average brain score of each network and (2) its performance or its training step (Fig. 4 and Supplementary Fig. 1). Positive and negative correlations indicate convergence and divergence, respectively. Brain scores above 0 before training indicate a fortuitous relationship between the activations of the brain and those of the networks.

**Permutation feature importance**. To systematically quantify how the architecture, language accuracy, and training of the language transformers impacted their ability to linearly map onto brain activity, we fitted, for each subject separately, a Random Forest across the models' properties to predict their brain scores, using scikit-learn's `RandomForest`[48,49]. Specifically, we input the following features to the random forest: the training task (causal language modeling "CLM" vs. masked language modeling "MLM"), the number of attention heads $\in$ [4, 8], the total number of layers $\in$ [4, 8, 12], dimensionality $\in$ [128, 256, 512], training step (number of gradient updates, $\in$ [0, 4.5M]), language modeling accuracy (top-1 accuracy at predicting a masked word) and the relative position of the representation (a.k.a "layer position", between 0 for the word-embedding layer, and 1 for the last layer). The performance of the Random Forest was evaluated for each subject separately with a Pearson correlation $R$ using five-split cross-validation across models.

"Permutation feature importance" summarizes how each of the covarying properties of the models (their task, architecture, etc.) specifically impacts the brain scores[48]. Permutation feature importance was implemented with scikit-learn[49] and is summarized with $\Delta R$: the decrease in $R$ when shuffling one feature (using 50 repetitions). For each subject, we reported the average decrease across the cross-validation splits (Fig. 4). The resulting scores ($\Delta R$) are expected to be centered around 0 if the corresponding feature does not impact the brain scores, and positive otherwise.

---

**Table 1 Brain parcellation Taxonomy used to label the regions of interest in the brain following the PALS Brodmann's Area atlas[88].**

| Label | Corresponding Brodmann's areas |
|---|---|
| V1 | BA17 |
| Fusiform | BA37 |
| Angular | BA39 |
| aSTG | BA22-anterior |
| mSTG | BA22-middle |
| pSTG | BA22-posterior |
| Supramarginal | BA40 |
| Infero-frontal | BA44/BA45/BA47 |
| Fronto-polar | BA10 |
| Temporo-polar | BA38 |

**Statistics and reproducibility**. To estimate the robustness of our results, we systematically performed second-level analyses across subjects. Specifically, we applied Wilcoxon signed-rank tests across subjects' estimates to evaluate whether the effect under consideration was systematically different from the chance level. The $p$-values of individual voxel/source/time samples were corrected for multiple comparisons, using a False Discovery Rate (Benjamini/Hochberg) as implemented in MNE-Python[92] (we use the default parameters). Error bars and ± refer to the standard error of the mean (SEM) interval across subjects.

**Brain parcellation**. In Fig. 3, we focus on particular regions of interest using the Brodmann's areas from the PALS parcellation of freesurfer[86]. The superior temporal gyrus (BA22) is split into its anterior, middle and posterior parts to increase granularity. For clarity, we rename certain areas as specified in Table 1.

**Ethics**. These data were provided (in part) by the Donders Institute for Brain, Cognition, and Behavior after having been approved by the local ethics committee (CMO—the local "Committee on Research Involving Human Subjects" in the Arnhem-Nijmegen region). As stated in the original paper[37], "In the informed consent procedure, [the subjects] explicitly consented for the anonymized collected data to be used for research purposes by other researchers. [..] The study was approved by the local ethics committee (CMO—the local "Committee on Research Involving Human Subjects" in the Arnhem-Nijmegen region) and followed guidelines of the Helsinki declaration."

**Reporting summary**. Further information on research design is available in the Nature Research Reporting Summary linked to this article.

## Data availability
The data are publicly available on request. They were provided by the Donders Institute for Brain, Cognition and Behavior after having been approved by the local ethics committee (CMO—the local "Committee on Research Involving Human Subjects" in the Arnhem-Nijmegen region). Link: https://data.donders.ru.nl/collections/di/dccn/DSC_3011020.09_236. The aggregated data used to generate Fig. 2 (Supplementary Data 1), Fig. 4 (Supplementary Data 2), and Fig. 3 (Supplementary Data 3) are available jointly with the manuscript. In particular, to generate Fig. 3, one needs Supplementary Data 3a (scores across layers for MEG, Fig. 3a), Supplementary Data 3b (scores across training for MEG, Figure Fig. 3b), Supplementary Data 3c (permutation importance for MEG, Fig. 3e), Supplementary Data 3d (scores across layers for fMRI, Fig. 3e), Supplementary Data 3e (scores across training for fMRI, Fig. 3f), and Supplementary Data 3f (permutation importance for fMRI, Fig. 3h).

## Code availability
The code is available upon request. Data analysis was performed in Python using the scikit-learn open source library[49]. The MEG and fMRI data were processed using MNE-Python[92], nilearn[91] and freesurfer[86]. The natural language processing algorithms were trained using the implementation from the XLM github repository (https://github.com/facebookresearch/XLM,[6]).

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

## Acknowledgements

This work was supported by ANR-17-EURE-0017, the Fyssen Foundation, and the Bettencourt and Fyssen Foundations to J.R.K. for his work at PSL.

## Author contributions

J.R.K. defined the line of research, C.C. conducted the analyses, both authors analyzed the results, designed the figures and wrote the paper.

## Competing interests
The authors declare no competing interests.
