## [Peer Review File · Communications Biology]

Reviewers' comments:

Reviewer #1 (Remarks to the Author):

The authors competently present a clear and concise collection of results that lead them to three major conclusions. First, they argue that transformer language models converge to brain-like representations through training. Second, they determine that, among all tested properties of the language models, language model performance (i.e. how well it can do the task it is trained to do; predict missing words) is the best predictor of how well a model matches the brain. Third, they show that the language model representations are most predictive of brain responses at a later time (1+ seconds) than other, lower-level representations.

This paper is very technically strong: the analyses are well-motivated, clearly explained, and seem well-executed. (I do have one minor technical question, but it is not at all critical so I will save for the end of this review.) It is also well-written, timely, and interesting. I am going to spend a little time criticizing a couple points that I think could be improved, but I want to be clear that I think that this is, overall, a Good Paper!

But while this paper is very technically strong, there are, I believe, some small weaknesses in how far the authors go in drawing conclusions from the data. In particular, I think the authors are actually under-selling what are some very interesting results.

The core claim of the paper is that better language models are also better models of language processing in the brain. But what, really, does it mean for models to "converge to brain-like representations during their training"? A vexatious linguist might argue that this apparent convergence is merely a side-effect of the model learning about various linguistic phenomena (for example: part of speech, or syntactic structure). As the model is trained, it gets better at language modeling by learning these phenomena, and its ability with these phenomena is what causes it to be more similar to the brain. Surely this disproves this nonsense about language models being brain-like. "Yes," the authors should answer in exasperation, "we showed that is true in Figure S6. It is entirely consistent with our claims."

To avoid this annoying criticism entirely, I encourage the authors to spill a little more ink on the question of what language model representations are. What information might these representations contain? What does it mean for the representations to contain information that is also represented in the brain, or information that is not represented in the brain? Also, I encourage the authors to highlight the (very) interesting results from Figure S6, which show that language model performance, brain score, and the ability of language model representations to do other tasks are all linked.

Minor points:

* The authors use the adjective "compositional" to describe the LM-derived embeddings (line 121, etc.). I am not completely opposed to this, but I think it would be more accurate to use the broader term "contextual" or "contextualized", which the authors do earlier in the paper (line 23) and in the methods (line 236). "Compositional" implies a very specific type of information related to combinations of words or concepts within syntactic structures. While the LM-derived embeddings almost certainly DO contain compositional information, they also contain other information that is not compositional. For example, LM representations do an excellent job at disambiguating word sense using context—to call this "composition" would be a stretch.

* The very high prediction performance of both the word and contextual embedding models in retinotopic visual cortex (even after controlling for low-level visual appearance using the visual CNN model!) really looks artificial. The authors suggest that this may be due to feedback from other brain areas, but that just doesn't square with other results in the field, particularly those that use longer narrative stimuli. This weird result is by no means a deal-breaker — keeping it or removing it doesn't change anything about the core claims of the paper — but its weirdness will definitely give some readers pause. I'd encourage the authors to look into other means by which visual information (or even oculomotor behavior?) might be present in the language embeddings. For example, is the contextual embedding better able to represent sentence start/end than the

other embeddings?

Reviewer #2 (Remarks to the Author):

In this paper, Caucheteux & King examine the ability of transformer neural networks trained on word prediction tasks to fit representations in the human brain measured with fMRI and MEG. The authors show that in the middle layers of the transformers the representations can be used to predict brain activity at levels close to the noise ceiling. Furthermore, the ability to predict brain activity is strongly correlated with the models' abilities to solve the word prediction tasks. They go on to show that the time-course of this predictability for visual, lexical, and syntactic/semantic representations matches the expected role of different brain regions in processing linguistic stimuli.

Altogether, I think this is a great paper and very worthy of publication. I don't actually have any major concerns with it. There are a series of smaller things that I think the authors can and should attend to, which I detail here:

- Lines 54-55: any correction for multiple comparisons? Not stated here, but methods suggest FDR. Should be mentioned here in passing at least.
- Line 58: would be good to explain to the reader what CLM and MLM are here (not all readers would know this, and it's frustrating as a reader to have to check the methods for a key piece of info like this).
- Lines 78-79: if you have 32 transformers, and 100 learning steps, how do you arrive at 32,400 word embeddings? The numbers don't make sense as described here.
- Lines 81-82: Missing a left bracket somewhere...
- Line 118-120: given that the embedding are word-by-word, how could there possibly be sub-lexical features? Aren't those necessarily dropped by the word embeddings? So, why control for this? The result is still interesting, but the way it's phrased (as a control) is odd. Really, it's that you want to compare a visual embedding to a linguistic embedding, it's not a control, per se.
- Figure 3 legend: what are MEG gains exactly? The legend suggests it's just the R for the lexical and compositional models, minus the R for the visual model. But, if that is the case, how is there a blue trace on these plots and what does it represent?
- Lines 198-200: it should be noted that a similar phenomenon is seen in ResNet models of vision trained on image categorization (see e.g. <https://www.biorxiv.org/content/10.1101/407007v2.full.pdf>).
- Lines 200-202: What does it mean to "generate the meaning of a sentence"? It is not clear to me that this is a well-posed loss, nor a relevant concept for systems neuroscience. An alternative, more concrete explanation for this finding is that the brain is doing more than just language tasks, and thus, the over-specialization is a result of there being other losses for other purposes in the brain, unlike in these transformer models. I think this is a more comprehensible explanation (since "meaning" is itself an ill-defined term), and so should be considered in the text. Plus, the brain likely does try to predict words from context like these transformers! Humans can certainly do that... Moreover, there's growing evidence that prediction is one of the core loss functions shaping computation in the neocortex (see for example: <https://www.sciencedirect.com/science/article/pii/S0896627318308572>). It is probably worth recognizing that, given that the results here add further support to the idea that the neocortex is engaged in predictive learning.
- Lines 204-205: It is probably worth noting that transformers may actually be implementing a set of computations that can be performed with recurrent neural networks. See: <https://arxiv.org/abs/2008.02217>

- Lines 247-251: did you do any hyperparameter optimization? Need to clarify.
- Supp. Table 1: Are these just the noise ceilings for fMRI? Why not report those for MEG too?

Reviewer #3 (Remarks to the Author):

This is a high-quality paper. It describes a set of analyses where the authors linearly map between neural network models of language and two neuroimaging datasets (fMRI and MEG). The key novel contributions of the paper are (a) using both MEG and fMRI and (b) systematically varying the factors that might affect neural network performance (architecture, the size of the training set).

Here are some suggestions about how the paper can be improved.

1. Framing. The current framing doesn't quite do justice to the existing literature. A large body of work relating language models and brain data are called "preliminary findings" (p.1). Moreover, the result that's presented as the main novel contribution of the paper — a correlation between model language performance and brain score — has previously been shown by Schrimpf et al, 2020 (Figure 3). This paper's contribution is still worthwhile because their analyses are more controlled and detailed, but the fact that this key result has been demonstrated before (even if tentatively) needs to be acknowledged.
2. Word embedding layer. I'm having some trouble interpreting the word embedding results. I must be missing something but shouldn't the word embeddings remain the same throughout the training process, since they're not part of the model being trained? If these results are about the first layer, then it might need to be renamed accordingly.
3. Low correlation. The low correlation value certainly makes the results more questionable. The authors spend a good amount of time explaining why the values might be so low and why the results are still interesting, and I'm willing to accept this line of reasoning. I think it might be helpful to plot the null permutation distribution (in addition to reporting the significance) for a better demonstration of the result. It might also help to plot the distribution of correlation values across channels/voxels to show that, in the language regions, correlation values are indeed substantially higher.
4. Noise ceiling. I think it's misleading to call the between-subject prediction measure the noise ceiling. It's possible that there are individual differences in how subjects process sentences, but these differences can plausibly be captured by a language model (e.g., one subject would have increased activation to the word "cat", and that can be accounted for in the mapping). So, the measure is not a "ceiling" for model performance, nor does it measure noise (but rather variability between subjects).
5. Interpreting the mapping. What does it mean to have a transformer layer linearly map onto an fMRI/MEG recording? The authors discuss this question a bit toward the end of the paper, but what exactly does it tell us about possible computational/representational parallels between brains and ANNs? And why use a linear mapping and not, say, RSA, for the most veridical comparison? These are hard questions but discussing them a little more would help.
6. Orthogonalizing the predictors. In the methods, the authors mention the option of orthogonalizing the visual, word-level, and contextualized embeddings in order to isolate their contributions. This is an analysis I was also going to suggest. I don't understand why the authors don't do it. Just because the gain is positive, doesn't mean that we wouldn't want to quantify the independent contributions of each embedding.
7. Feature importance. Is there a way to definitively show that model architecture features aren't simply used as proxies for model performance? I understand that, when the architectural feature is shuffled, model performance scores remain, but I wonder if some deeper correlational structure is being exploited.

Specific points:

- Axis tick missing in Figure 2B; impossible to evaluate the scale.
- In Figure 3D, please show raw scores and not gain — those are more informative.
- Lines 80-83: why are these scores much higher than those reported above? (e.g. in Figure 2)

- Lines 85-86: for each of the three layers, there should be 2 values, MEG and fMRI, but only 1 value is reported in parentheses 2 and 3.
- Line 119: you say the CNN is trained on character recognition but in the methods you specify that it's trained on word recognition from images, which makes much more sense given the aims of the analysis. Please rephrase.
- Lines 203-210: this paragraph seems unfinished. If the models are different from the brain, how should it affect our interpretation of the results?
- Lines 220-221: "deep language networks can be used as meaningful models of language processing only if they process language like our brain." This sentence is confusing given the paragraph above about the divergences between transformer models and the brain, as well as the fact that language performance and brain score start to diverge at some point.

We thank the reviewers for their very constructive comments.

All three reviewers praised the clarity and quality of the manuscript:

- R1 says that our “well-written, timely, and interesting” paper is “very technically strong, well-motivated, clearly explained” and “well-executed”, and overall, a “Good Paper”.
- R2 and R3 add that our “high-quality” (R3) and “great” (R2) paper is “very worthy of publication” (R2).

All three reviewers suggested “minor” modifications to improve the manuscript and clarify “small” points. In particular, R1 advised that we emphasize “very interesting results”, potentially “undersold”. Thus, in addition to small technical clarifications, we have

- Polished the figures and clarified the structures of the results and discussion.
- Amended the abstract and slightly modified the title (from “The mapping of deep language models on brain responses primarily depends on their performance” to “The similarity between deep language algorithms and the brain primarily depends on their ability to predict words from context”).
- Modified the discussion to (1) emphasize the implications of the results and (2) precise the definitions of “compositionality” and “representation”.

As suggested by R3, we also added new analyses to strengthen the robustness of the results (orthogonalization method, permutation distribution as a baseline score, and distribution of scores across voxels, cf. Figure S8, S9, S10 and S11).

We are grateful to our reviewers for helping improve the quality and readability of our manuscript.

Reviewer #1 (Remarks to the Author):

The authors competently present a clear and concise collection of results that lead them to three major conclusions. First, they argue that transformer language models converge to brain-like representations through training. Second, they determine that, among all tested properties of the language models, language model performance (i.e. how well it can do the task it is trained to do; predict missing words) is the best predictor of how well a model matches the brain. Third, they show that the language model representations are most predictive of brain responses at a later time (1+ seconds) than other, lower-level representations.

This paper is very technically strong: the analyses are well-motivated, clearly explained, and seem well-executed. (I do have one minor technical question, but it is not at all critical so I will save for the end of this review.) It is also well-written, timely, and interesting. I am going to spend a little time criticizing a couple points that I think could be improved, but I want to be clear that I think that this is, overall, a Good Paper!

But while this paper is very technically strong, there are, I believe, some small weaknesses in how far the authors go in drawing conclusions from the data. In particular, I think the authors are actually under-selling what are some very interesting results.

The core claim of the paper is that better language models are also better models of language processing in the brain. But what, really, does it mean for models to “converge to brain-like representations during their training”? A vexatious linguist might argue that this apparent convergence is merely a side-effect of the model learning about various linguistic phenomena (for example: part of speech, or syntactic structure). As the model is trained, it gets better at language modeling by learning these phenomena, and its ability with these phenomena is what causes it to be more similar to the brain. Surely this disproves this nonsense about language models being brain-like. “Yes,” the authors should answer in exasperation, “we showed that is true in Figure S6. It is entirely consistent with our claims.”

To avoid this annoying criticism entirely, I encourage the authors to spill a little more ink on the question of what language model representations are. What information might these representations contain? What does it mean for the representations to contain information that is also represented in the brain, or information that is not represented in the brain? Also, I encourage the authors to highlight the (very) interesting results from Figure S6, which show that language model performance, brain score, and the ability of language model representations to do other tasks are all linked.

We thank Reviewer 1 for this engaging remark, to which we can only but agree. We now add a discussion paragraph to clarify this general issue, and the implications of the convergence in particular.

"Second, our study shows that the similarity between deep language models and the brain primarily depends on their ability to predict words from their context. Specifically, we show that language performance is the most contributing factor explaining the variability of brain scores across embeddings (Figure 2D and H). Analogous results have been reported in both vision and audition research, where best deep learning models tend to best map onto brain responses (di Carlo & Yamins 2014, Yamins et al 2016, Schrimpf et al 2018, Schrimpf et al 2020, Kell et al 2018, Millet & King 2021). Together, these results suggest that deep learning algorithms “converge” – at least partially – to brain-like representations during their training. This result is not trivial: the representations that are optimal to predict masked or future words from large amounts of texts could have been very distinct from those the brain learns to generate.
[...]

At this stage, however, these three levels representations remain coarsely defined. Further inspection of artificial (Manning et al 2020, Lakretz et al 2019) and biological networks (Caucheteux et al 2021, Reddy et al 2020, Hale et al 2021) remains necessary to further decompose them into interpretable features. In particular, it will be important to test whether the “converging” representations presently identified solely correspond to well-known linguistics phenomena as our supplementary analyses suggest (e.g. Figure S6), or, on the contrary, whether they correspond to unknown language structures.”

Minor points:

* The authors use the adjective “compositional” to describe the LM-derived embeddings (line 121, etc.). I am not completely opposed to this, but I think it would be more accurate to use the broader term “contextual” or “contextualized”, which the authors do earlier in the paper (line 23) and in the methods (line 236). “Compositional” implies a very specific type of information related to combinations of words or concepts within syntactic structures. While the LM-derived embeddings almost certainly DO contain compositional information, they also contain other information that is not compositional. For example, LM representations do an excellent job at disambiguating word sense using context—to call this “composition” would be a stretch.

Thank you for this remark. To clarify this issue, we refer the reader to the following discussion:

“Following a recently proposed taxonomy (Caucheteux, Gramfort & King 2021), we formally define “compositional” as the language representations that cannot be explained by the linear combination of lexical representations.

This definition may not be fully aligned with the many definitions of compositionality proposed over the years (Szabó, 2004). Specifically, some linguists restrict compositionality to the limited, generally invertible, combinations of words that follow the laws of syntax, and would consequently thus prefer the term “contextual”. We believe, however, that the latter term does not clearly point to the representations that are more than the sum of their parts (Frege, 1884, Pelletier & Jeffry, 1994) which is critical to the present analyses (Section 2.4, Figure 3).”

* The very high prediction performance of both the word and contextual embedding models in retinotopic visual cortex (even after controlling for low-level visual appearance using the visual CNN model!) really looks artificial. The authors suggest that this may be due to

feedback from other brain areas, but that just doesn't square with other results in the field, particularly those that use longer narrative stimuli. This weird result is by no means a deal-breaker – keeping it or removing it doesn't change anything about the core claims of the paper – but its weirdness will definitely give some readers pause. I'd encourage the authors to look into other means by which visual information (or even oculomotor behavior?) might be present in the language embeddings. For example, is the contextual embedding better able to represent sentence start/end than the other embeddings?

We agree with Reviewer 1 that this finding is not expected. R1 is right that contextual embeddings *do* encode information about the position of words in the sentence as well as the beginning and end of the sentence (e.g. "The" and "Once" are more likely to be at the beginning of a sentence than "pretty"). This may create a bias in this dataset, in which sentences are separated by clear pauses. Critically, however, the *specific* contribution of contextual embeddings compared to the visual embedding are particularly small in V1 (Figure 3D). We now add a paragraph in the result's section to clarify this point:

"Finally, lexical and compositional embeddings accurately predict brain responses in the early visual cortex. This result is not necessarily surprising: language embeddings encode features (e.g. position of words in the sentence, beginning/end of the sentence) that correlate with visual information (words are flashed at a screen, and the sentences are separated by pauses). Critically, the gain (ΔR) of these embeddings remain very small, suggesting that this effect is mainly driven by the covariance between low- and high-level representations of words."

Reviewer #2 (Remarks to the Author):

In this paper, Caucheteux & King examine the ability of transformer neural networks trained on word prediction tasks to fit representations in the human brain measured with fMRI and MEG. The authors show that in the middle layers of the transformers the representations can be used to predict brain activity at levels close to the noise ceiling. Furthermore, the ability to predict brain activity is strongly correlated with the models' abilities to solve the word prediction tasks. They go on to show that the time-course of this predictability for visual, lexical, and syntactic/semantic representations matches the expected role of different brain regions in processing linguistic stimuli.

Altogether, I think this is a great paper and very worthy of publication. I don't actually have any major concerns with it. There are a series of smaller things that I think the authors can and should attend to, which I detail here:

- Lines 54-55: any correction for multiple comparisons? Not stated here, but methods suggest FDR. Should be mentioned here in passing at least.

We thank Reviewer 2 for identifying this missing information, and now specify in the manuscript:

“Finally, we assess the statistical significance of these (average or single-voxel/channel) brain scores with a two-sided Wilcoxon test across subjects, after testing for multiple comparisons across voxels using False Discovery Rate (cf. Methods 4.5).*

**We use the FDR implementation of the MNE package using default parameters (35).”*

- Line 58: would be good to explain to the reader what CLM and MLM are here (not all readers would know this, and it’s frustrating as a reader to have to check the methods for a key piece of info like this).

We thank Reviewer 2 for this remark, and now indicate:

“What computational principle leads these deep language algorithms to generate brain-like activations? To address this issue, we generalize the above analyses and evaluate the brain scores of 32 transformer architectures (varying from 4 to 12 layers, each ranging from 128 to 512 dimensions, and each benefiting from 4 to 8 attention heads), trained on the same Wikipedia dataset either with a ‘causal’ language modeling (CLM) or a ‘masked’ language modeling task (MLM). While causal language models are trained to predict a word from its previous context, masked language models are trained to predict a randomly masked word from its both left and right context.”

- Lines 78-79: if you have 32 transformers, and 100 learning steps, how do you arrive at 32,400 word embeddings? The numbers don’t make sense as described here.

We thank Reviewer 2 for this remark. We agree that our number can be confusing, given the complexity of our combinatorial analysis. We now add a paragraph in the Methods’ section to clarify this point:

“For clarity, we dissociate:

1) The architectures (e.g one transformer with 12 layers): there are 32 transformer architectures here (16 CLM and 16 MLM).

2) The models: one architecture, frozen at one particular learning step. Since we use 100 learning steps, there are $32 \times 100 = 3,200$ networks here.

3) The *embeddings*: one word representation extracted from a network, at one particular layer. Since the number of layers varies with the architecture (twelve networks with 5, twelve networks with 9 and twelve networks with 13 twelve layers, including the non contextualized word embedding), there are $12 \times (5 + 9 + 13) = 324$ representations per step, so $324 \times 100 = 3,400$ word embeddings in total."

- Lines 81-82: Missing a left bracket somewhere...

Corrected, thank you.

- Line 118-120: given that the embedding are word-by-word, how could there possibly be sub-lexical features? Aren't those necessarily dropped by the word embeddings? So, why control for this? The result is still interesting, but the way it's phrased (as a control) is odd. Really, it's that you want to compare a visual embedding to a linguistic embedding, it's not a control, per se.

Word embedding may encode certain features that correlate with sub-lexical and visual information. For instance, word embeddings encode the word frequency, which is highly correlated with the word length. Nonetheless, we agree that the word "control" may not be optimal and corrected the text and figures accordingly: e.g.

"We also report the brain scores of a convolutional neural network trained on visual character recognition (blue) to account for low-level visual representations"

- Figure 3 legend: what are MEG gains exactly? The legend suggests it's just the R for the lexical and compositional models, minus the R for the visual model. But, if that is the case, how is there a blue trace on these plots and what does it represent?

We thank Reviewer 2 for this remark and have modified the legend to clarify this issue (R2 was misguided because the blue lines were not properly defined in our original manuscript!):

"B. Mean MEG scores averaged across all time samples and subjects. C. Left: mean MEG scores averaged across all sensors. Right: mean MEG gains averaged across all sensors: i.e. the gain in MEG score of one level relative to the level below (blue: $R[\text{visual}]$; green: $R[\text{word}] - R[\text{visual}]$; red: $R[\text{compositional}] - R[\text{word}]$)."

- Lines 198-200: it should be noted that a similar phenomenon is seen in ResNet models of vision trained on image categorization (see e.g. <https://www.biorxiv.org/content/10.1101/407007v2.full.pdf>).

Thank you for this reference, we now corrected:

“Specifically, we show that language performance is the most contributing factor explaining the variability of brain scores across embeddings (Figure 2D and H). Analogous results have been reported in both vision and audition research, where best deep learning models tend to best map onto brain responses (di Carlo & Yamins 2014, Yamins 2016, Schrimpf et al 2018, Schrimpf et al 2020, Kell et al 2018, Millet and King 2021). Together, these results suggest that deep learning algorithms “converge” -- at least partially -- to brain-like representations during their training.

- Lines 200-202: What does it mean to “generate the meaning of a sentence”? It is not clear to me that this is a well-posed loss, nor a relevant concept for systems neuroscience. An alternative, more concrete explanation for this finding is that the brain is doing more than just language tasks, and thus, the over-specialization is a result of there being other losses for other purposes in the brain, unlike in these transformer models. I think this is a more comprehensible explanation (since “meaning” is itself an ill-defined term), and so should be considered in the text. Plus, the brain likely does try to predict words from context like these transformers! Humans can certainly do that... Moreover, there’s growing evidence that prediction is one of the core loss functions shaping computation in the neocortex (see for example: <https://www.sciencedirect.com/science/article/pii/S0896627318308572>). It is probably worth recognizing that, given that the results here add further support to the idea that the neocortex is engaged in predictive learning.

We completely agree with R2: “the meaning of a sentence” is not well posed and the objective function(s) of the brain remain(s) a major unknown to system neuroscience. To discuss this issue, we now rephrase the paragraph as follows:

“The conclusion that deep networks converge towards brain-like representations should be qualified: we show that the brain scores of the very best models tend to ultimately decrease with language performance, especially in fMRI (Figure 2G). We speculate that this phenomenon (also observed in vision Schrimpf et al 2018) may arise because transformers overfit an inappropriate objective. Specifically, while there is growing evidence that the human brain does predict words from context (Keller et al 2018, Heilbron et al 2020, Goldstein et al 2021), this learning rule may not fully account for the complex (and potentially various) tasks performed by the brain (e.g. long-range (Wang et al. 2020, Lee et al. 2021) and hierarchical predictions (Friston, 2010)).

- Lines 204-205: It is probably worth noting that transformers may actually be implementing a set of computations that can be performed with recurrent neural networks. See: <https://arxiv.org/abs/2008.02217>

We agree and thus added:

“On the other hand, transformers are i) feedforward neural networks, [...]*

**Note that, given large-enough spaces, feedforward transformers may actually implement computations similar to recurrent networks (Ramsauer et al. 2020).”*

- Lines 247-251: did you do any hyperparameter optimization? Need to clarify.

We now clarify:

“The algorithms were trained using XLM implementation, using the default hyperparameters”*

**[...]. No hyper-parameter tuning was performed.*

- Supp. Table 1: Are these just the noise ceilings for fMRI? Why not report those for MEG too?

We report the MEG “noise ceilings” in Figure S4. Given the high temporal variability of these estimates, we thought that reporting a full table of [regions x time] would be less readable.

Reviewer #3 (Remarks to the Author):

This is a high-quality paper. It describes a set of analyses where the authors linearly map between neural network models of language and two neuroimaging datasets (fMRI and MEG). The key novel contributions of the paper are (a) using both MEG and fMRI and (b) systematically varying the factors that might affect neural network performance (architecture, the size of the training set).

Here are some suggestions about how the paper can be improved.

1. Framing. The current framing doesn't quite do justice to the existing literature. A large body of work relating language models and brain data are called “preliminary findings” (p.1). Moreover, the result that's presented as the main novel contribution of the paper – a correlation between model language performance and brain score – has previously been

shown by Schrimpf et al, 2020 (Figure 3). This paper's contribution is still worthwhile because their analyses are more controlled and detailed, but the fact that this key result has been demonstrated before (even if tentatively) needs to be acknowledged.

In the original text, we had indicated: *"Do these algorithms process words and sentences like the human brain? Preliminary evidence suggests that they might. First, word embeddings -- high dimensional dense vectors trained to predict lexical neighborhood (Bengio et al. 2001, Mikolov et al. 2013, Pennington et al. 2014, Bojanowski et al. 2016) -- have been shown to linearly map onto the brain responses elicited by words presented either in isolation (Mitchell et al. 2008, Anderson et al. 2019, Sassenhagen et al. 2019) or within narratives (Oota et al. 2018, Abnar et al. 2019, Ruan et al. 2016, Brodbeck et al. 2018, Gauthier et al. 2018, Wehbe et al. 2014)."*

While cited elsewhere in the text, we agree that this sentence should also cite Schrimpf et al. 2020, and thus amended the text accordingly.

We would like to point out that Schrimpf's et al. study was actually submitted on bioRxiv the same week as the present study did, and remains, to date, not published. Our manuscript systematically received one extremely negative review in our past submissions, which explains why the delay of the present submission.

Schrimpf et al s results, based on 15 fMRI and 5 ecog subjects (vs 100 fMRI and 95 MEG subjects here), focuses on (1) the subselection of the 10% best voxels in each subject (as opposed to whole brain, and full dynamics here) and (2) on a few pretrained models provided by Huggingface which vary along multiple dimensions (corpora, dimensionality, objectives, performance etc) -- as opposed to a systematic grid-search that allows us to distinguish the specific contributions of architecture, training, objectives and performance.

We thus believe that it is neither fair nor accurate to say that "the result that's presented as the main novel contribution of the paper -- a correlation between model language performance and brain score -- has previously been shown by Schrimpf et al, 2020"

Supplementary Figure: A summary of the neuroimaging and electrophysiological studies mapping deep language models onto brain responses to sentences. Italic designates preprint papers up to 2020. Our study stands out as investing many more models in many more subjects using both fMRI and MEG, and can thus confidently 1) compare models with one-another and 2) provide remarkably detailed brain maps and dynamics of each of the stages of language processing.

2. Word embedding layer. I'm having some trouble interpreting the word embedding results. I must be missing something but shouldn't the word embeddings remain the same throughout the training process, since they're not part of the model being trained? If these results are about the first layer, then it might need to be renamed accordingly.

Here, we decided not to report pre-trained word embeddings (e.g. Glove, Word2Vec), but to solely report the word embedding layer of the transformers (i.e. the layer that maps each token to a vector). This non-contextual layer is simultaneously trained with the other layers of the network. Thus, the weights of the word embedding vary throughout the training.

Our internal analyses suggest that Glove and Word2Vec show very similar patterns to these word embeddings. However, because the training of these models requires a different approach and amounts of data, we opted for simplicity.

We now clarify this issue.

"Transformers" consist of multiple contextual "transformer layers" stacked onto one non-contextualized word embedding layer (a look-up table).*

**Following the standard implementation (Vaswani et al. 2017, Devlin et al. 2019, Radford et al., 2020), the word embedding layer is trained simultaneously with the*

contextual layers. Thus, the weights of the word embedding vary with the training, and so do their activations in response to fixed inputs.”

3. Low correlation. The low correlation value certainly makes the results more questionable. The authors spend a good amount of time explaining why the values might be so low and why the results are still interesting, and I’m willing to accept this line of reasoning. I think it might be helpful to plot the null permutation distribution (in addition to reporting the significance) for a better demonstration of the result.

We agree and now add the null permutation distribution. As described below, we compute the brain score between the embeddings and the fMRI of the subjects, shuffled across time samples. The resulting permutation scores are close to zero.

Figure S8: Permutation distribution. As a baseline, we compare the normal R scores (dark colors) to those of a permutation distribution (light colors) for each of the visual (blue), lexical (green) and compositional embeddings (red) introduced in Figure 3. For each (subject, voxel) pair, we compute the mapping between the embeddings X and the fMRI of the subject, either (i) shuffled across time samples or (ii) without shuffling. Above, we report scores averaged across subjects and voxels. Error bars are standard-error of the mean across subjects.

It might also help to plot the distribution of correlation values across channels/voxels to show that, in the language regions, correlation values are indeed substantially higher.

We agree, and now added a histogram of brain scores across voxels (fMRI) and channels (MEG).

Figure S9: Distribution of R scores across fMRI voxels (left) and MEG sources (right). We compute the brain scores for the visual (blue), lexical (green) and compositional (red) embeddings introduced in Figure 3. We average scores across voxels (resp. sources) and subjects, to obtain one single score per voxel (resp. source). Above, the corresponding distribution of the R scores across voxels and sources.

4. Noise ceiling. I think it’s misleading to call the between-subject prediction measure the noise ceiling. It’s possible that there are individual differences in how subjects process sentences, but these differences can plausibly be captured by a language model (e.g., one subject would have increased activation to the word “cat”, and that can be accounted for in the mapping). So, the measure is not a “ceiling” for model performance, nor does it measure noise (but rather variability between subjects).

We agree with Reviewer 3. The term “noise ceiling” can be misleading because its corresponding analyses always make an assumption on how to measure the upper bound of the signal-to-noise ratio. Yet, this assumption varies across studies. For example, Huth et al. 2016 assume that presenting twice the same podcast to a subject will lead to twice the same signal in the fMRI scanner. This assumption is false, however, because brain responses to words vary between repetitions, especially in high-level cortical regions (Dehaene-Lambertz et al., 2006). Others, like Richard et al. 2019 assume that temporal time course but not the spatial pattern of a fMRI response to a stimulus is identical across subjects. Again, this assumption is false because we know that different individuals present different brain dynamics (Friederici, 2017). In this sense, our “noise ceiling” is also incorrectly judging the true level of noise: i.e. we assume that subjects’ fMRI responses are linearly explainable by the population-average brain response to the same stimulus.

Together, these elements confirm R3’s concern, i.e. what is generally referred to as “noise ceiling” in our domain is misleading. We thus amended the manuscript as follows:

“To quantify the proportion of these brain responses that depend on the specific content of sentences, we fit, for each subject separately, a shared response model across subjects (a.k.a. “noise-ceiling”, see Methods, Figure 2B-D). “

5. Interpreting the mapping. What does it mean to have a transformer layer linearly map onto an fMRI/MEG recording? The authors discuss this question a bit toward the end of the paper, but what exactly does it tell us about possible computational/representational parallels between brains and ANNs?

Thanks for raising this issue. We now indicate in the discussion:

“The notion of representation underlying this mapping is formally defined as “linearly-readable information”. This operational definition helps identify brain responses that any neuron can differentiate -- as opposed to “entangled information” which would necessitate several layers before being usable (Marvin Minsky and Seymour Papert, 1969, Cadieu et al. 2014, Kriegeskorte et al. 2008, King & Dehaene, 2014; Cohen, 2020)”.

And why use a linear mapping and not, say, RSA, for the most veridical comparison? These are hard questions but discussing them a little more would help.

Representational similarity analysis (RSA) has been shown to be largely equivalent to linear encoding (Diedrichsen & Kriegeskorte, PLoS computational biology, 2017). However, RSA is suited for discrete and repeated trials (e.g. 32 visual categories, each repeated 20 times). In speech, however, 1) the stimulation is a continuous flow and 2) repeating a stimulus (e.g. a sentence) does not evoke the same neural responses in each brain region (Dehaene-Lambertz et al HBM 2006). Consequently, encoding single occurrences seems best suited to compare the activations of brains and deep neural networks.

6. Orthogonalizing the predictors. In the methods, the authors mention the option of orthogonalizing the visual, word-level, and contextualized embeddings in order to isolate their contributions. This is an analysis I was also going to suggest. I don't understand why the authors don't do it. Just because the gain is positive, doesn't mean that we wouldn't want to quantify the independent contributions of each embedding.

This is a good point. Our subtractive approach is, in principle, more conservative than an a priori orthogonalization because it assumes that all of the shared variance between two hierarchical models (e.g. word embedding and contextualized embedding) is fully attributed to the lowest-level. i.e. the neural bases of “compositional/contextual” representations that we evidence here cannot be explained by the lower-level models presently tested.

However, we agree that this is an interesting control, and added the results of this analysis in Figure S10.

Figure S10: Comparison between two orthogonalization methods. In Figure 3, we report the raw brain scores (without subtraction) for the visual (blue, X_V), lexical (green, X_W) and compositional (red, X_C) embeddings (“base method” on the left). On the right, for each level, we subtract the scores of the level below (e.g. red scores $R_C = \mathcal{R}(X_C) - \mathcal{R}(X_W)$). In the middle, we orthogonalize the predictors before computing the brain scores, by “regressing out” the effect of the lower level onto the current level. For the compositional score R_C , we fit a ridge regression model* f to predict X_C given the concatenation of the visual and word embeddings $X_V \oplus X_W$. Then, we compute the brain scores of the residuals $\tilde{X}_C = X_C - \hat{f}(X_V \oplus X_W)$. We proceed similarly for the lexical residuals $\tilde{X}_W = X_W - X_V$. As we see, the subtraction method (right) is more conservative than the method with regress out (middle).

*We use the RidgeCV implementation from scikit-learn, with 10 possible penalization values log spaced between 10^{-3} and 10^8 .

7. Feature importance. Is there a way to definitively show that model architecture features aren't simply used as proxies for model performance? I understand that, when the architectural feature is shuffled, model performance scores remain, but I wonder if some deeper correlational structure is being exploited.

Feature importance is input with both architectural and performance properties. To maximize the specificity of our analysis, we used a Random Forest (Breiman et al. 2001), i.e. a model known to efficiently capture both linear and nonlinear correlation structures. We would be happy to consider alternative models instead.

Specific points:

- Axis tick missing in Figure 2B; impossible to evaluate the scale.

We have now modified the figure.

- In Figure 3D, please show raw scores and not gain — those are more informative.

We now added this figure to supplementary Figure S11. Note that this is precisely the reverse request that we received in previous submissions.

Figure S11: Same as Figure 3C and D, but without subtracting the scores of the level below.

- Lines 80-83: why are these scores much higher than those reported above? (e.g. in Figure 2)

Thank you very much for pointing this out, this value, attributed to randomly initialized networks, was actually incorrect. We now amended the manuscript, and the conclusions remain unchanged.

- Lines 85-86: for each of the three layers, there should be 2 values, MEG and fMRI, but only 1 value is reported in parentheses 2 and 3.

Thank you for pointing this out. We now added the two missing values.

- Line 119: you say the CNN is trained on character recognition but in the methods you specify that it's trained on word recognition from images, which makes much more sense given the aims of the analysis. Please rephrase.

Thank you for pointing this out, we agree that this statement is unclear. We now indicate:

“Specifically, this model was trained on real pictures of single words taken in naturalistic settings (e.g. ad, banner).”

And give the link to the corresponding model:

<https://github.com/clovaai/deep-text-recognition-benchmark>.

- Lines 203-210: this paragraph seems unfinished. If the models are different from the brain, how should it affect our interpretation of the results?

Thank you for pointing this out, we corrected the discussion accordingly.

- Lines 220-221: “deep language networks can be used as meaningful models of language processing only if they process language like our brain.” This sentence is confusing given the paragraph above about the divergences between transformer models and the brain, as well as the fact that language performance and brain score start to diverge at some point.

Thank you for pointing this out, we re-structured the discussion to clarify this issue.

REVIEWERS' COMMENTS:

Reviewer #2 (Remarks to the Author):

The authors have addressed my concerns, and I think the paper is good for publication.

Reviewer #3 (Remarks to the Author):

The authors did a thorough job addressing the reviews; I only have a couple minor comments left.

Title (and terminology): The phrase "deep language algorithms" isn't a commonly used term in the field (2 google search results) and might be confusing to the readers. I'd recommend "deep language models" but leaving this up to the authors.

Re the literature review: I was mainly concerned with the phrase "preliminary evidence" given that there is a reasonably large number of works in that area (as cited by the authors). A slight rewording should suffice.

Re Schrimpf et al: If the papers were indeed preprinted at the same time, it is a fair point that their result does not precede yours. However, the fact that a version of this finding is shown in another paper should still be acknowledged given that it is now part of the literature. As I mentioned, your demonstration is much more convincing because of its thoroughness, but saying "our results are consistent with preliminary findings by Schrimpf et al (reported around the same time as ours), who..." would be really helpful to the reader and would provide an accurate representation of the state of the field.

I still think that showing the raw scores in Figure 3D is more informative (the interpretation is more intuitive for most people & it'd be easier to compare with results from other papers), but given that the reviewers' opinions vary on this point, this is up to the authors.

Given its length, we propose to modify the title and to use from now on: "Brains and algorithms partially converge in natural language processing".

Reviewer #3 (Remarks to the Author):

The authors did a thorough job addressing the reviews; I only have a couple minor comments left.

Title (and terminology): The phrase "deep language algorithms" isn't a commonly used term in the field (2 google search results) and might be confusing to the readers. I'd recommend "deep language models" but leaving this up to the authors.

We agree and no longer use the phrase "deep language algorithms".

Re the literature review: I was mainly concerned with the phrase "preliminary evidence" given that there is a reasonably large number of works in that area (as cited by the authors). A slight rewording should suffice.

We now use the phrase "recent work suggests [...]".

Re Schrimpf et al: If the papers were indeed preprinted at the same time, it is a fair point that their result does not precede yours. However, the fact that a version of this finding is shown in another paper should still be acknowledged given that it is now part of the literature. As I mentioned, your demonstration is much more convincing because of its thoroughness, but saying "our results are consistent with preliminary findings by Schrimpf et al (reported around the same time as ours), who..." would be really helpful to the reader and would provide an accurate representation of the state of the field.

We agree and now indicate: "Our results are consistent with the findings of Schrimpf et al. reported simultaneously to ours."

I still think that showing the raw scores in Figure 3D is more informative (the interpretation is more intuitive for most people & it'd be easier to compare with results from other papers), but given that the reviewers' opinions vary on this point, this is up to the authors.

We agree that the raw scores are informative. However, we believe that the interpretation is clearer with the subtraction. We thus keep the raw scores in Supplementary Figure S5.